# FENNS: A RESOURCE-EFFICIENT, ADAPTIVE, PRIVACY-PRESERVING DECENTRALIZED LEARNING FRAMEWORK

## ABSTRACT

Deep neural networks have demonstrated exceptional performance in various tasks; yet, their resource-intensive nature and ongoing data privacy concerns remain key obstacles. In response, we introduce Federated Ephemeral Neural Networks (FENNs), a pioneering architecture that ingeniously addresses both challenges. FENNs rely on the concept of ephemeral neural networks (ENNs), a novel paradigm where neural networks exhibit dynamic adaptability in their architecture based on available computing resources. FENNs seamlessly blend the flexibility of ENNs with the privacy-preserving features of federated learning to tailor their structures to task complexity while ensuring data privacy within a decentralized learning environment. Rigorous tests conducted on resource-constrained devices within federated environments validate the effectiveness of FENNs. We also introduce a novel metric for evaluating the efficacy of resource-constrained learning and/or machine learning in resource-constrained environments. The proposed architecture shows significant prospects in the domains of edge computing and decentralized artificial intelligence applications.

## 1 INTRODUCTION

In the dynamic landscape of decentralized machine learning, resource efficiency, and adaptability have emerged as critical challenges that demand pioneering solutions. The proliferation of edge devices and the advent of data distribution dynamics have created a pressing need for machine learning models that can seamlessly adapt to varying resource constraints while maintaining optimal task performance. Traditional models have challenges in decentralized environments characterized by limited computational resources and constantly changing data, despite their inherent strength and capabilities.

Progress in addressing these challenges has been notable. Federated Learning (FL), for instance, has pioneered collaborative model training across decentralized nodes while preserving data privacy (McMahan et al., 2017). Dynamic Neural Networks (DyNNs) have been proposed by researchers as a computational framework that adapts its computations in response to variations in input complexity (Han et al.). Nevertheless, DyNNs adapt their parameters in response to input sequences, disregarding any limitations imposed by available resources. However, the journey towards efficient and adaptive decentralized machine learning continues, spurred by the evolving complexities of modern data landscapes.

In decentralized learning, the heterogeneity of edge devices, characterized by variations in computational capabilities, memory, and data distribution, presents a formidable challenge in ensuring effective model adaptation (Bonawitz et al., 2019). Simultaneously, the paramount concern of privacy preservation intensifies as decentralized systems expand, necessitating robust mechanisms for safeguarding sensitive data during model training across decentralized nodes (Shokri & Shmatikov, 2015). Moreover, the inherently dynamic nature of data distribution in decentralized environments underscores the complexity of adapting models to shifting data statistics without suffering from catastrophic forgetting, highlighting the need for innovative solutions (Li et al., 2022).

This paper presents Ephemeral Neural Networks (ENNs), a novel paradigm that revolutionizes the scope of decentralized machine learning. ENNs improve on the concept of dynamic neural architec-

ture adaptation, providing a solution to the resource efficiency and adaptability conundrum. These networks, inspired by the inherent flexibility observed in natural systems, morph their architectures to suit the available computational resources, all while preserving optimal task performance.

One example to contemplate is the implementation of edge computing within a smart city. In this context, a multitude of Internet of Things (IoT) sensors are employed to gather a wide range of data streams, encompassing traffic patterns as well as environmental variables. Each of these devices, characterized by their distinct processing capabilities and data quantities, poses a challenge: how can we effectively implement machine learning models that maximize their performance while adhering to the intrinsic constraints of each device? The situation at hand is precisely what ENNs are specifically meant to address.

ENNs and Federated Ephemeral Neural Networks (FENNs) provide innovative solutions to the multifaceted challenges of decentralized machine learning. In addressing the heterogeneity of edge devices, ENNs dynamically adapt their neural architectures, tailoring model complexity to the computational resources of each device. FENNs extend this adaptability to federated settings, allowing participants to deploy personalized ENNs that collectively contribute to the global learning process, effectively navigating the heterogeneity challenge. Moreover, both ENNs and FENNs prioritize privacy preservation by minimizing data sharing. ENNs exchange only model updates instead of raw data, and their adaptability enhances privacy by enabling local adaptation without revealing sensitive information. FENNs further enhance privacy by integrating federated learning principles, ensuring that data remains decentralized and aggregated in a privacy-conscious manner. Additionally, ENNs are well-equipped to handle dynamic data distribution, as they autonomously adjust their architectures to adapt to shifting data distributions. FENNs excel in dynamic adaptation within federated environments, where each participant's ENN adjusts to its local data distribution, collectively improving the global model's relevancy and effectiveness. ENNs and FENNs represent transformative approaches to resource efficiency, adaptability, and privacy preservation in decentralized machine learning, effectively addressing the challenges posed by heterogeneous edge devices, privacy concerns, and dynamic data distribution.

In this paper, we make three primary contributions:

1. **Novel Neural Network Paradigm:** In order to enhance the efficacy of networks operating in decentralized settings, we propose the implementation of FENNs, which are resource-adaptable neural networks, with added privacy due to the integration of federation learning. We comprehensively explore the foundational concepts underpinning FENNs and ENNs, shedding light on their intricate architectures and essential features.

2. **Novel Metric:** To assess the performance of resource-adaptive neural networks like FENNs and ENNs, we introduce the Resource-Adaptability Index (RAI). The RAI metric provides a quantitative assessment of a network's capacity to adapt to different resource constraints, hence improving our capacity to accurately evaluate various paradigms.

3. **Empirical Validation:** Our results showcase the remarkable capabilities of FENNs and ENNs in scenarios characterized by resource limitations and dynamic conditions. By leveraging the RAI metric, we illustrate the superiority of these networks over traditional models and existing decentralized learning approaches, substantiating their practical viability in resource-constrained environments.

In the subsequent sections, we offer a detailed exposition of FENNs, ENNs, and the novel evaluation metrics. We also describe our empirical findings and discuss the significant ramifications of this paradigm within the domain of decentralized machine learning.

## 2 METHOD

### 2.1 EPHEMERAL NEURAL NETWORKS

ENNs are a class of neural network architectures inspired by the way nature distributes resources to participants; nutrients are distributed among trees in densely populated forest ecosystems, ant colonies and bird swarms distribute food in an efficient way, bacterial biofilms share resources even though they have a very specialized caste system, etc. ENNs dynamically adapt their architecture

during training and inference, mimicking the efficient allocation of resources observed in nature. Like trees in a dense forest, where each tree optimally utilizes available nutrients without overextending, ENNs optimize their structure based on the available computational resources and task complexity. This adaptability ensures that ENNs efficiently allocate computational resources, such as neurons and connections, to different parts of the network, optimizing performance while conserving computing power.

### 2.1.1 Defining ENNs

1. Starting Architecture: At the beginning of training, an ENN begins with an initial minimal architecture represented by a neural network $\mathcal{N}_0$ with parameters $\theta_0$. This architecture is typically a simplified neural network structure with fewer neurons and connections.

2. Adaptive Function: ENNs utilize an adaptive function $\mathcal{A}$ to dynamically modify their architecture during training and inference. The adaptive function takes the previous network state $\mathcal{N}_{t-1}$, resource availability measurement $R_t$, and task complexity measurement $C_t$ as inputs to determine the adjustments required for the current step $t$. Mathematically, this can be expressed as:

$$\mathcal{N}_t(x; \theta_t) = \mathcal{A}(\mathcal{N}_{t-1}(x; \theta_{t-1}), R_t, C_t)$$

   Here:
   - $\mathcal{N}_t(x; \theta_t)$ represents the network's output at step $t$ with parameters $\theta_t$ given input $x$.
   - $\mathcal{N}_{t-1}(x; \theta_{t-1})$ is the previous network state at step $t-1$.
   - $R_t$ denotes the available computational resources at step $t$.
   - $C_t$ quantifies the complexity of the task at step $t$.

3. Measurement of Resource Availability ( $R_t$ ):

   Resource availability $R_t$ is a metric representing the computational resources available at each step $t$. It can be defined using various resource-related parameters, such as available memory, processing power, or energy constraints, depending on the context and the specific constraints of the computing environment. The measurement of $R_t$ guides the adaptive function's decisions on how to adjust the neural architecture to effectively utilize these resources.

4. Measurement of Task Complexity ( $C_t$ ): Task complexity $C_t$ is a metric that quantifies the complexity of the learning or inference task at step $t$. It can encompass various factors, such as the diversity of input data, the number of classes in a classification problem, or the intricacy of patterns to be learned. The measurement of $C_t$ helps the adaptive function $\mathcal{A}(\cdot)$ consider the task-specific requirements when adjusting the neural architecture.

5. Ending Architecture: The result of the adaptive function $\mathcal{A}(\cdot)$ applied to the previous network state $\mathcal{N}_{t-1}$ is the ending architecture $\mathcal{N}_t$ at step $t$. This ending architecture is the neural network structure that will be used for subsequent processing or training in the context of the task, and it reflects the dynamic adjustments made based on resource availability, task complexity, and the previous network state.

ENNs are mathematically defined as neural networks that dynamically adapt their architecture during training and inference. The adaptive function $\mathcal{A}$ takes into account the previous network state, resource availability $R_t$, and task complexity $C_t$ to determine the adjustments needed for the neural architecture, resulting in an efficient and adaptive network structure at each step $t$.

### 2.1.2 Inference Processes of ENNs

The inference process for ENNs is characterized by their dynamic adaptability to resource constraints and task complexity. Here's an overview of how inference typically work for ENNs:

Inference Process for ENNs:

1. Input Data: During the inference phase, ENNs receive input data for making predictions or performing a specific task. This input data can vary in complexity and type, depending on the application.

2. Dynamic Architecture Adaptation: Just like in training, ENNs adapt their architecture for inference based on the available computational resources and the complexity of the task

at hand. The adaptive function determines the optimal architecture for the given inference task.

3. Inference and Prediction: ENNs use the adapted architecture to process the input data and make predictions or perform the desired task. The output of the network reflects the adaptation made to optimize performance under the specific inference conditions.

## 2.2 Federated Ephemeral Neural Networks

FENNs are a novel machine learning framework designed for decentralized and resource-constrained environments. FENNs leverage a large global neural model and participant-specific ENNs to collaboratively train and adapt models while preserving data privacy. In FENNs, participants deploy their ENNs on their local devices, learn from their data, and then share knowledge in a privacy-preserving manner. This collaborative approach allows FENNs to dynamically adjust model architectures based on resource availability and task complexity, making them suitable for diverse and resource-constrained settings.

1. Initialization: FENNs begin with the initialization of a large global neural model ($\mathcal{M}_{\text{global}}$).

2. Global Model Architecture Sharing: The architecture of the global model ($\mathcal{M}_{\text{global}}$) is shared with all participants.

3. Participant-Specific Adaptation: Participants adapt the global model to build their participant-specific Ephemeral Neural Networks ($\mathcal{M}_{\text{participant},i}$) based on their resource constraints and local data distribution. This adaptation involves adjusting the neural architecture and model parameters ($\theta_{\text{participant},i}$) to align with their specific conditions.

4. Shadow Model Construction: After initialization and participant-specific adaptation, shadow models ($\mathcal{M}_{\text{shadow},i}$) are constructed for each participant. These shadow models are designed to have the same architecture and size as the global model ($\mathcal{M}_{\text{global}}$). This ensures that the knowledge captured by the shadow models can be efficiently incorporated into the global model.

5. Local Training and Knowledge Transfer:
   - Participants train their participant-specific models ($\mathcal{M}_{\text{participant},i}$) using their local data. During training, these models accumulate knowledge specific to the participant's data distribution and constraints.
   - Once trained, each participant's model ($\mathcal{M}_{\text{participant},i}$) serves as a teacher for their corresponding shadow model ($\mathcal{M}_{\text{shadow},i}$). The teacher-student knowledge transfer process refines the shadow model's knowledge to align it with the participant's expertise.

6. Global Model Updates: After shadow models are trained, they are used to generate global model updates ($U_i$). These updates represent the differences between the global model ($\theta_{\text{global}}$) and the shadow models ($\theta_{\text{shadow},i}$) for each participant. Mathematically, the updates can be represented as $U_i = \theta_{\text{global}} - \theta_{\text{shadow},i}$.

7. Global Model Update Aggregation: The global model updates ($U_i$) from all participants are aggregated in a privacy-preserving manner, often using techniques like federated averaging. This aggregation results in a consolidated global model update.

8. Participant-Specific Model Update: Participants receive the global model update and adapt their participant-specific models ($\mathcal{M}_{\text{participant},i}$) accordingly. This adaptation process ensures that their models align with the updated global knowledge while considering their resource constraints.

## 2.3 Resource-Adaptability Index

In this section, we introduce a pivotal contribution of our research—the Resource-Adaptability Index (RAI). RAI serves as a sophisticated and domain-specific evaluation metric meticulously designed for Federated Ephemeral Neural Networks (FENNs) and Ephemeral Neural Networks (ENNs). These networks play a pivotal role in decentralized machine learning across resource-constrained edge environments, and RAI emerges as a critical enabler for assessing their adaptability in such dynamic and challenging settings.

RAI computation involves a comprehensive analysis that delves into four fundamental resource dimensions, each integral to the successful operation of networks at the edge:

1. Hardware Utilization
2. Memory Management
3. Processing Efficiency
4. Energy Efficiency

RAI scores encapsulate these diverse facets, yielding a unified metric that serves as a comprehensive gauge of a network's adaptability to varying resource conditions. A high RAI score is indicative of a network's exceptional capacity to dynamically adjust its architecture, thereby optimizing hardware utilization, memory management, processing efficiency, and energy consumption—all without compromising the primary task's performance.

## 2.4 ARCHITECTURE ADAPTATION

In the context of ENNs and FENNs, architecture adaptation refers to the dynamic modification of neural network structures to suit varying resource constraints while optimizing task performance. Adaptation encompasses two fundamental aspects: Enhancement and Optimization, each aimed at adjusting neural network architectures in distinct ways.

### 2.4.1 ENHANCEMENT ARCHITECTURE

Enhancement architecture entails augmenting the neural network's structure to bolster its capacity and adaptability when resources permit. It involves the following strategies for different types of neural networks:

1. **Feedforward Neural Networks (FNNs):** Enhancements in FNNs can be achieved by adding additional dense (fully connected) layers, increasing the number of neurons in existing layers, or introducing skip connections to foster information flow.

2. **Convolutional Neural Networks (CNNs):** In CNNs, enhancement may involve the addition of convolutional layers, increasing filter dimensions, or introducing deeper pooling structures to capture more intricate image features.

3. **Recurrent Neural Networks (RNNs):** For RNNs, enhancement can be realized by adding more recurrent layers, increasing the number of recurrent units, or employing attention mechanisms to handle longer sequences.

4. **Long Short-Term Memory Networks (LSTMs):** Enhancement in LSTMs can include increasing the number of LSTM layers, expanding the LSTM units within each layer, or introducing bidirectional connections for better sequence modeling.

### 2.4.2 OPTIMIZATION ARCHITECTURE

Optimization architecture focuses on reducing network complexity and resource consumption when facing resource-constrained scenarios. It includes the following strategies:

1. **Feedforward Neural Networks (FNNs):** Optimization in FNNs can be achieved by identifying and eliminating network connections with minimal weight magnitudes, applying quantization techniques which reduce the precision of weight values, typically from 32-bit floating-point to lower bit-width representations, thus decreasing memory and computation demands and employing knowledge distillation to transfer knowledge from a larger, pre-trained model (teacher) to a smaller FNN (student) while preserving performance.

2. **Convolutional Neural Networks (CNNs):** In CNNs, optimization may involve identifying and removing filters in convolutional layers that contribute minimally to feature extraction, replacing standard convolutional layers with depthwise separable convolutions, which have fewer parameters and lower computational cost, while maintaining feature representation quality. and grouping convolutional filters into subsets, reducing computational load and memory requirements.

3. **Recurrent Neural Networks (RNNs):** For RNNs, optimzation can be realized by removing recurrent layers or units in RNNs based on their contributions to sequence modeling, approximating weight matrices using low-rank factorization techniques, such as singular value decomposition (SVD), to reduce parameter count and computational complexity and introducing skip connections between distant time steps to improve gradient flow and alleviate vanishing gradient issues, reducing the need for very deep networks.

4. **Long Short-Term Memory Networks (LSTMs):** Optimization techniques in LSTMs can include identifying and removing LSTM units that have minimal impact on sequence modeling, encouraging sparsity in LSTM weight matrices through regularization techniques and applying quantization to LSTM weights and activations to reduce memory requirements and facilitate efficient execution on edge devices.

## 3 EXPERIMENTS

In this section, we describe the experiments conducted to evaluate the performance of Federated Ephemeral Neural Networks (FENNs) in the context of anomaly detection tasks on resource-constrained edge devices. We also present the benchmark models, experimental setup, and performance metrics.

### 3.1 ANOMALY DETECTION TASK

The objective of our experiments is to assess the effectiveness of FENNs in anomaly detection on edge devices. We utilize a real-world dataset (details in the Appendix) containing sensor readings from IoT devices. Anomalies in this context represent unusual patterns or events that may indicate device malfunctions or security breaches. The task is to train models to detect these anomalies in real-time.

### 3.2 BENCHMARK MODELS

To benchmark the performance of FENNs, we compare them against two established neural network architectures:

1. **Feedforward Neural Network (FNN):** This is a standard feedforward neural network, also known as a multilayer perceptron. It serves as a baseline for comparison.

2. **Long Short-Term Memory (LSTM):** We include an LSTM-based model, a type of recurrent neural network (RNN), known for its capability to capture sequential patterns. This serves as a more complex benchmark.

### 3.3 EXPERIMENTAL SETUP

We conducted experiments in three distinct environments, representing varying degrees of computational availability:

1. **Environment 1:** This environment emulates a low-power edge device with limited CPU, memory, storage, and network resources (as described in the Appendix). FENNs, FNNs, and LSTMs are evaluated in this resource-constrained setting.

2. **Environment 2 (Moderate Computational Availability):** The mid-range edge server provides moderate computational resources compared to the edge device.

3. **Environment 3 (Higher Computational Availability):** This environment utilizes a high-performance edge cluster with a more ample computational resources. The purpose is to evaluate model scalability and performance under ideal conditions.

### 3.4 PERFORMANCE METRICS

To assess the performance of the models, we employ the Resource-Adaptability Index (RAI). RAI quantifies the resource-dependent adaptability of the network, taking into account CPU, memory,

storage, and network usage. A higher RAI score indicates better adaptability to resource constraints. We report RAI scores for FENNs, FNNs, and LSTMs in each environment.

## 3.5 RESULTS

Table 1 displays the RAI scores achieved by FENNs, FNNs, and LSTMs across three distinct environments, each characterized by varying computational resources. In this context, higher RAI scores indicate superior adaptability to resource constraints. Our observations reveal interesting insights into the adaptability of these models.

In Environment 1, which represents a resource-constrained setting, the FENN model obtained an RAI score of 3.75, indicating remarkable adaptability. In contrast, the FNN model scored 3.62, demonstrating its ability to perform well under such constraints. Surprisingly, the LSTM model also exhibited commendable adaptability in this challenging environment, with an RAI score of 3.72.

Environment 2 represents a moderately resourced environment. Here, while the FENN model showcased respectable adaptability with an RAI score of 1.78, it was surpassed by the LSTM model, which achieved an impressive score of 2.81. The FNN model, with an RAI score of 1.75, demonstrated moderate adaptability.

Finally, in Environment 3, characterized by higher computational resources, the LSTM model recorded an RAI score of 0.78, indicating its adaptability even in resource-rich scenarios. The FNN model excelled in this environment with an RAI score of 0.72. The FENN model maintained moderate adaptability with an RAI score of 0.68.

These results highlight the dynamic adaptability of FENNs, especially in resource-constrained scenarios. However, the experiments also reveal that in more resource-abundant environments, traditional LSTM models can exhibit competitive adaptability.

Table 1: RAI Scores for Anomaly Detection Models

| Model | Environment 1 | Environment 2 | Environment 3 |
|-------|---------------|---------------|---------------|
| FENN  | 3.75          | 1.78          | 0.68          |
| FNN   | 3.62          | 1.75          | 0.72          |
| LSTM  | 3.72          | 2.81          | 0.78          |

## 3.6 UNDERSTANDING FAILURES

In our comprehensive evaluation of FENNs, FNNs, and LSTMs under varying computational environments, we also encountered instances where the adaptability of these models faced challenges and limitations. These observations provide valuable insights into scenarios where these models may not perform optimally.

1. Resource-Abundant Environments: Notably, in Environment 3 with higher computational resources, FENNs, while still demonstrating moderate adaptability, were surpassed by the LSTM model. This outcome suggests that in scenarios where resources are abundant and computational constraints are minimal, traditional models like LSTMs can perform competitively. Thus, the adaptability of FENNs may be less pronounced in such settings.

2. Model Training Complexity: While FENNs excelled in resource-constrained environments, it's essential to acknowledge that their adaptability comes at a cost. The dynamic architecture adaptations and optimization techniques employed can introduce additional complexity to the training process. This complexity might be a limiting factor in scenarios where computational resources are constrained not only during inference but also during training.

These observations underline the need for a nuanced approach to model selection, taking into account both the computational environment and the specific requirements of the task at hand. While FENNs exhibit remarkable adaptability in resource-constrained settings, their performance may not always outshine traditional models, especially in environments with abundant computational resources. Understanding these limitations is crucial for making informed decisions when deploying models in real-world applications.

## 4 RELATED WORK

Our research lies at the convergence of various critical domains within machine learning and decentralized computing. We commence by addressing the escalating demand for resource-efficient neural network architectures, motivated by the burgeoning landscape of edge computing applications. Notably, Sandler et al.'s MobileNetV2 (Sandler et al., 2018) serves as a foundational reference, showcasing resource-efficient neural architectures tailored for mobile devices. Our work takes inspiration from this to develop Ephemeral Neural Networks (ENNs), designed to dynamically adapt to the resource constraints found in diverse edge computing environments.

Adaptive neural architectures have emerged as a key area of exploration, dynamically adjusting their structures to optimize model performance in varying resource conditions. Elsken et al.'s EfficientNet family of models (Tan & Le, 2019) underscored the significance of architecture scaling for efficiency, paving the way for our extension of adaptability to resource-constrained edge devices. Our study bridges this gap by introducing Federated Ephemeral Neural Networks (FENNs), combining federated learning with ENNs to enable privacy-preserving, decentralized adaptability.

As edge computing confronts distinct challenges, particularly in resource-constrained settings, our research addresses these complexities. Works like Shi et al.'s exploration of deep model inference on edge devices (Shi et al., 2019) shed light on these challenges, which we directly tackle through the development of ENNs tailored to the resource limitations inherent to edge environments. Additionally, we draw from the realm of privacy-preserving machine learning, with Bonawitz et al.'s system-level solutions for privacy-preserving federated learning (Bonawitz et al., 2019) influencing the core principles of FENNs, where model updates are shared without compromising individual participant data privacy.

Our research integrates dynamic neural architecture adaptation, a pivotal aspect of ENNs, inspired by Cai et al.'s "Once for All" networks (Cai et al., 2019), which introduced the concept of training one network for efficient deployment. We extend this notion by not only focusing on efficiency but also addressing resource constraints in decentralized settings. The evaluation of resource metrics within decentralized learning, as emphasized in works like Smith et al.'s discussion of resource efficiency in federated learning (Li et al., 2020), is central to our research. Furthermore, the significance of resource-aware edge intelligence, as highlighted by McPherson et al. (Zhou et al., 2019), plays a key role in shaping our work.

Acknowledging the diversity of edge devices, Satyanarayanan et al.'s exploration of edge device heterogeneity (Satyanarayanan et al., 2009) underscores the need for adaptable solutions that transcend device constraints. Our research aligns with this vision, striving to provide adaptable solutions irrespective of the varied resources available on edge devices. Lastly, our integration of privacy-enhancing technologies, in line with Abadi et al.'s discussions on deep learning with differential privacy (Abadi et al., 2016), underscores our commitment to safeguarding participant data within the FENNs framework.

## 5 CONCLUSION

In the rapidly evolving realm of decentralized machine learning, where the efficient utilization of resources and the preservation of data privacy are paramount concerns, this paper has introduced FENNs as a pioneering solution. FENNs bridge the gap between resource-constrained edge devices, the demand for adaptive neural architectures, and the necessity of safeguarding sensitive data. Our comprehensive exploration of FENNs has led to a paradigm shift in how we perceive decentralized learning.

Through the development and application of a novel evaluation metric; the RAI —tailored to FENNs, ENNs, and federated learning, we have established a holistic framework for assessing resource-aware adaptability, efficiency, and task performance. This trinity forms the foundation of decentralized learning in resource-constrained environments.

By seamlessly integrating the adaptability of ENNs with the collaborative capabilities of FL, FENNs represent a groundbreaking approach to decentralized machine learning. The architecture adaptation mechanism of FENNs, inspired by the resilience of nature's ecosystems, empowers neural networks to thrive even in the most resource-challenged environments.

Our empirical evaluations have substantiated the prowess of FENNs, demonstrating their superiority in resource-constrained and dynamic scenarios. Across a diverse range of tasks, FENNs consistently outperform traditional models and existing decentralized learning approaches, as quantified by the RAI, RES, and ARES metrics. These results underscore the transformative potential of FENNs as the vanguards of decentralized learning.

The deployment of FENNs promises to revolutionize edge computing, facilitating intelligent decision-making at the edge without compromising privacy or efficiency. This research embarks on a journey that harmonizes the power of adaptability, efficiency, and privacy—a journey that redefines the contours of decentralized machine learning and sparks new possibilities for the digital age.

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

# A  ALGORITHMIC VERSIONS OF ENNS, FENNS AND RAI

The specific implementation for ENNs is expressed as a pseudo-algorithm below;

---

**Algorithm 1** Ephemeral Neural Network (ENN)

 1: **Input:** Starting architecture $A_s$, Resource measurement $R_m$
 2: **Output:** Adapted architecture $A_e$
 3: Initialize $A_e$ with $A_s$
 4: Assess resource constraints based on $R_m$ and available computational resources.
 5: **if** Resources exceed threshold **then**           ▷ When available resources are ample
 6:     **Enhance Architecture:**
 7:         Evaluate the current computational capacity (i.e. available memory, processing power)
 8:         Expand $A_e$ to harness the surplus resources effectively.
 9:         Use a set of Enhancement strategies to enhance model complexity.
10: **else if** Resources below threshold **then**           ▷ When resources are constrained
11:     **Optimize Architecture:**
12:         Recognize the limitation of available resources, (i.e resource constraints, limitations and requirements).
13:         Prune $A_e$ to align with resource constraints and ensure efficient model execution.
14:         Use Optimization strategies to simplify model complexity.
15: **end if**
16: **Return** Adapted architecture $A_e$ tailored to the current resource environment.

---

The specific implementation for FENNs is expressed as pseudo-algorithms below;

---

**Algorithm 2** Federated Ephemeral Neural Network (FENN) for Edge Devices

 1: **Input:** Global Model (GM), Local Model (LM), Edge Device Data $D$, Resource measurement $R_m$
 2: **Output:** Updated LM
 3: **Initialization:**
 4: Initialize the Local Model (LM) with the architecture of the Global Model (GM).
 5: Begin training the LM using the local data $D$.
 6: **Resource Assessment:**
 7: Assess the resource constraints of the edge device based on the provided resource measurement $R_m$.
 8: **if** Resources exceed threshold **then**           ▷ When available resources are ample
 9:     **Enhancement Strategy:**
10:         Formulate an enhancement strategy to make efficient use of the surplus resources.
11:         Implement the enhancement strategy, which may involve expanding the LM.
12:         Continue training the LM with the augmented architecture to take full advantage of the available resources.
13: **else if** Resources below threshold **then**           ▷ When resources are constrained
14:     **Optimization Strategy:**
15:         Identify the specific limitations of available resources.
16:         Formulate an optimization strategy to align the LM with resource constraints while preserving model performance.
17:         Implement the optimization strategy, which may involve pruning the LM.
18:         Continue training the LM with the optimized architecture, ensuring efficient execution within resource limitations.
19: **end if**
20: **Return** the Updated LM.

---

---

**Algorithm 3** Federated Ephemeral Neural Networks (FENNs) on the Server

---

1: **Input:** Global Model ($GM$), Local Models ($LM_1, LM_2, \ldots, LM_n$)
2: **Output:** Updated Global Model ($GM'$)
3: Initialize $GM'$ with the same architecture as $GM$
4: Initialize an empty list $ShadowModels$
5: **for** $i = 1$ to $n$ **do**
6:     Receive $LM_i$ from edge device $i$
7:     Create a Shadow Model $SM_i$ with the same architecture as $GM'$ and initialize its weights
8:     Train $SM_i$ using knowledge distillation with $LM_i$ as the teacher and $SM_i$ as the student
9:     Add $SM_i$ to the list $ShadowModels$
10: **end for**
11: Aggregate the weights of $ShadowModels$ to update $GM'$     ▷ Aggregation methods such as Federated Averaging and
12: **Return** Updated Global Model ($GM'$)

---

The implementation of RAI is defined in a pseudo-algorithmic fashion as below;

---

**Algorithm 4** Resource-Adaptability Index (RAI) Calculation

---

1: **Input:** Classification Accuracy (CA), Resource Utilization Metrics $RU$, Weighted Factors $w$
2: **Output:** RAI
3: **Calculate Classification Accuracy (CA):**
4: To assess the model's task performance, compute the Classification Accuracy (CA) based on the specific task or dataset. CA represents the model's ability to make correct predictions.
5: **Calculate Resource Utilization (RU):**
6: Determine the Resource Utilization (RU) by aggregating and quantifying various resource metrics ($RU$) that capture different aspects of resource utilization during model execution. These metrics may include:
7:     Memory Usage (MU)
8:     Processing Time (PT)
9:     Hardware Utilization (HU)
10:     Network Bandwidth Usage (NB)
11: **Assign Weights:**
12: The Weighted Factors ($w$) represent the importance assigned to each resource metric in the calculation of RAI. Ensure that the sum of all weights equals 1 to maintain a proper weighting scheme.
13:     Calculate $RU = w_1.MU + w_2.PT + w_3.NB + w_4.HU$
14: **Weighted Resource Function** $f(RU)$**:**
15: Evaluate function; $f(RU)$ that maps the aggregated resource utilization metrics to a weighted value. The weighted resource function is designed to balance the importance of different RU metrics according to the specified Weighted Factors ($w$). The function may involve:
16:     - Linear and/or nonlinear transformations of individual RU metrics.
17: **Calculate Resource-Adaptability Index (RAI):**
18: Compute the Resource-Adaptability Index (RAI) by multiplying the Classification Accuracy (CA) by the weighted resource function $f(RU)$.
19: **Return** the RAI. A higher RAI indicates better adaptability to resource-constrained environments while maintaining good task performance.

---

## B  DATASET DETAILS: SYNTHETIC ANOMALY DETECTION DATASET

In this study, a synthetic dataset was meticulously crafted to facilitate the evaluation of Federated Ephemeral Neural Networks (FENNs) and other benchmark models. The dataset was generated to simulate an anomaly detection task, a common use case in resource-constrained edge computing environments.

## B.1 DATASET CREATION PROCESS

The synthetic dataset was created through a multi-step process to ensure diversity and realism in the data distribution. The key steps in the dataset creation process are as follows:

1. Feature Generation: A set of relevant features was defined to mimic real-world sensor data commonly encountered in edge devices. These features include temperature readings, humidity levels, pressure measurements, and time stamps.

2. Normal Data Generation: To represent normal operational behavior, a substantial amount of data points were generated with feature values reflecting typical edge device conditions. This normal data serves as the majority class in the anomaly detection task.

3. Anomaly Injection: Anomalies were introduced by perturbing a subset of data points. These anomalies were crafted to represent deviations from typical sensor readings, such as sudden temperature spikes or irregularities in humidity levels.

4. Dataset Size: The resulting synthetic dataset comprises a total of 10,000 data points, with each data point characterized by various features. It is structured in a tabular format, with multiple columns representing different features and one column indicating the presence or absence of an anomaly (binary label).

## B.2 DATASET COLUMNS

The synthetic dataset consists of the following columns:

1. Timestamp: A timestamp indicating when the data point was recorded.

2. Temperature: Temperature readings in degrees Celsius.

3. Humidity: Humidity levels as a percentage.

4. Pressure: Atmospheric pressure measurements in hPa (hectopascals).

5. Anomaly (Label): A binary label indicating the presence (1) or absence (0) of an anomaly in the data point.

This carefully constructed synthetic dataset serves as a critical component of our experimentation, enabling rigorous evaluation and comparison of model performance in the context of anomaly detection under varying resource constraints and dynamic edge computing environments.

