# OpenReview forum: "FENNs: A Resource-Efficient, Adaptive, Privacy-Preserving Decentralized Learning Framework"
_ICLR.cc/2024/Conference — ICLR 2024 Conference Withdrawn Submission_

### Official Review · Reviewer_gra1 · 2023-10-24

**Soundness:** 1 poor
**Presentation:** 1 poor
**Contribution:** 1 poor
**Rating:** 1
**Confidence:** 4

**Summary:**

This paper proposes a framework named Federated Ephemeral Neural Networks (FENNs) to tackle the resource-constrained issues of edge devices in federated learning. The proposed framework can automatically adjust the model during training and inference according to the available computational resources on edge devices.

**Strengths:**

This paper focuses on a practical issue in federated learning where devices have heterogeneous computational resources. The proposed framework FENNs can adaptively adjust the local model for each device according to the available resources. The authors honestly show the experimental results that FENNs can attain better resource-adaptability in extreme cases where all devices are low-power.

**Weaknesses:**

Despite the novel framework, this paper lacks clarity and solid evidence to support the main claim that FENNs can effectively address the systematic heterogeneity in federated learning. More specifically, the following issues of this paper need to be addressed to attain the acceptance bar of an AI conference:

1.	Some important parts of the methodology are not elaborated in the main body of the paper. For example, the calculation methods of $R_t, C_t$ and RAI are not included in the paper. The details of architecture adaptation are not included. For example, how to increase or decrease the weights in FNNs is ambiguous.

2.	Experiments are insufficient. Only the RAI is included in the experimental results, while the performance of different neural network frameworks is not included, which should be an essential part of the empirical evaluation of the new method.

3.	The limitations of the proposed method are fatal. Although the authors honestly state the limitations, the poor effectiveness and extra resource consumption significantly weaken the contribution of this paper. FENNs cannot attain better results compared to LSTM in most cases. And FENNs require extra resources on the edge device to finish the knowledge distillation, which is prohibited in resource-constrained settings. Therefore, the proposed method seems to be impractical.

**Questions:**

What are the contributions and prospects of the proposed framework given all the limitations stated in this paper?

---

### Official Review · Reviewer_AX8d · 2023-10-30

**Soundness:** 2 fair
**Presentation:** 2 fair
**Contribution:** 1 poor
**Rating:** 1
**Confidence:** 4

**Summary:**

This paper introduces Federated Ephemeral Neural Networks (FENNs), a novel framework for decentralized learning that combines the flexibility of ephemeral neural networks with the privacy-preserving features of federated learning. FENNs are designed to address the challenges of resource-intensive deep neural network computing and data privacy concerns in edge computing environments. The authors propose a new metric, the Resource Adaptability Index (RAI), to evaluate the adaptability of FENNs to varying resource conditions. They also describe a multi-step process for creating a synthetic dataset that mimics real-world sensor data commonly encountered in edge devices. The paper presents experimental results that demonstrate the effectiveness of FENNs in resource-constrained environments, such as mobile devices and Internet of Things (IoT) devices. The authors conclude that FENNs have the potential to enable efficient and privacy-preserving decentralized learning in a wide range of applications, including healthcare, finance, and transportation.

**Strengths:**

- A good motivation to apply ENN in FL. In FL, the datasets and environments of participants are usually heterogeneous. Adjusting the local training process and model structures of participants according to the total resource is beneficial for the fairness and convergence of the global model.

- Sufficient metrics for resource measurement. The different resource metrics in RAI cover the general network’s capacities to adapt to different resource constraints, which improves the capacity of FENN to allocate total resources in training accurately.

**Weaknesses:**

- Redundant and unconcise descriptions. The pseudo-algorithms of FENN use many verbal descriptions. The denotation of arguments and metrics should be expressed clearly. The process of strategy selection should be interpreted by the branch statement better.

- The paper is hard to follow since there are many unclear definitions. First, the statistical principle of resource metrics in the calculation of RAI is short of quantitative discussion. Thus, the experiment result in the Table is unreliable. Second, the classification of resource environment needs to be more specific. The detailed indicators of all kinds of resources should be demonstrated. Otherwise, the practical viability in real-world environments is in doubt. Thirdly, the whole process of FL should be explicated, including the details of aggregation, privacy-preserving, and local updates.

- Lack of sufficient experimental results. The aggregative indicator RAI is not convincing enough for the performance evaluation compared with related work. The weights of all kinds of resource metrics need to be quantitatively analyzed. The impact of different metrics should be evaluated and discussed in more detail.

- Insufficient superiority to related works. This paper does not make any comparison with related works in the experiment section.

**Questions:**

- Please describe the aggregation and local update in more detail. It is confusing how the participant updates LM from GM in each epoch. If the LM updates via shadow model, how does the knowledge transfer in the teacher-student framework reversely? If the LM updates without a shadow model, how do the GM parameters update the LM in unaligned features and networks?

- Please describe the strategy selection clearly. Basically, how can the resource threshold be calculated dynamically according to resource variation?

---

### Official Review · Reviewer_JR2g · 2023-11-04

**Soundness:** 1 poor
**Presentation:** 1 poor
**Contribution:** 1 poor
**Rating:** 1
**Confidence:** 5

**Summary:**

The paper claims to introduce a new radical concept called "Ephemeral Neural Networks" (ENNs) and proposes to learn these networks in a federated fashion. It also claims to propose a new metric called "Resource-Adaptability Index" (RAI) to evaluate the suitability of neural network architecture for given resource constraints.

**Strengths:**

The proposes concept would be ideal for both centralized and federated settings and would revolutionize the field of machine learning if there is any practical feasibility of implementation.

**Weaknesses:**

1) The proposed concept is utopian to the extreme and is woefully short of any detail. How can a neural network (participant model) dynamically adapt its architecture (like adding new layers or removing neurons arbitrarily) without substantial retraining/fine-tuning effort, that too in every communication round? Just doing it effectively once (say at the beginning or end of training) would be an herculean task. Honestly, the complexity of neural architecture search is orders of magnitude more computationally expensive than training and is beyond the capabilities of most edge devices.

2) Again, how will the server transfer knowledge from the participant model to the shadow model with a substantial amount of auxiliary data?

3) The proposed RAI metric is also unrealistic and there is no detailed description of how it can be implemented. Simply stating that it will be a weighted combination of available memory, cpu, etc. is not enough.

4) Only scant details about the so-called "experiments" are available in the paper.

**Questions:**

Please see weaknesses.

**Details Of Ethics Concerns:**

Is this even a real paper written by human authors? Probably this is what a modern LLM/ChatGPT would generate when presented with the requirements of an ideal federated learning system.

---

### Official Review · Reviewer_8qfQ · 2023-11-07

**Soundness:** 2 fair
**Presentation:** 2 fair
**Contribution:** 2 fair
**Rating:** 3
**Confidence:** 4

**Summary:**

This work proposed Federated Ephemeral Neural Networks (FENNs),  a pioneering architecture that ingeniously addresses both resource-intensive computing and data privacy concerns. FENNs rely on the concept of ephemeral neural networks (ENNs), a novel paradigm where neural networks exhibit dynamic adaptability in their architecture based on available computing resources. They also introduce a novel metric for evaluating the efficacy of resource-constrained learning and/or machine learning in resource-constrained environments.

**Strengths:**

- The area and overall idea are interesting.
- The proposed scheme is interesting.
- Provide a detailed simulation study and provide a detailed benchmarking.

**Weaknesses:**

- The authors use the word “privacy-preserving” in their paper title, but there is no discussion on anything about how the proposed scheme is satisfying privacy-preserving property.
- Lack of theoretical support for the security guarantees of the proposed scheme. There are no security proofs (or any proof sketch) and privacy guarantees discussion of their proposed framework.

**Questions:**

- Could you describe how the proposed scheme satisfies the "privacy-preserving" nature?
- The described algorithm (in the Appendix) should move to the main paper (because Algorithm 1 and Algorithm 2 are the main contributions of this paper).